# Post-stroke health-related quality of life following lower-extremity constraint-induced movement therapy - An observational survey study

Ingela Marklund [1,2*], Brynjar Fure[2,3], Maria Klässbo[2], Per Liv[4], Britt-Marie Stålnacke[1], Xiaolei Hu[1]

**1** Department of Community Medicine and Rehabilitation, Umeå University, Umeå, Sweden, **2** Centre for Clinical Research and Education, Region Värmland, Karlstad, Sweden, **3** School of Medical Sciences, Örebro University, Örebro, Sweden, **4** Department of Public Health and Clinical Medicine, Umeå University, Umeå, Sweden

\* ingela.marklund@umu.se (IM)

## Abstract

Lower- extremity constraint-induced movement therapy (LE-CIMT) has proven effective in overcoming physical disabilities. Participating in the LE-CIMT requires some independent walking ability without aids that indicates a higher level of motor function than for the entire stroke population. However, only few studies evaluated health-related quality of life (HRQoL) after LE-CIMT. This study aimed to compare HRQoL of people who had participated in LE-CIMT post-stroke to the general population and evaluate whether descriptive characteristics and clinical result were associated with their HRQoL. An observational survey study with a questionnaire including the Swedish RAND-36 and Saltin–Grimby Physical Activity Level Scale was sent to 162 people. Reference data from the Mid-Health Survey in Sweden was used for norm-based comparisons of RAND-36. Respondents' result from six-minute walk test post-LE-CIMT were used in the univariate analyse. The response rate was 65% ($n = 106$; 69 males and 37 females with a mean age of $62 \pm 12$ years). Ninety percent of the respondents could move around indoors and outdoors independently, despite this, 21% considered themselves physically inactive. The respondents had significantly reduced HRQoL compared to the general population in four of eight domains in the RAND-36: physical functioning ($p = 0.001$), role-functioning (physical; $p < 0.001$), general health ($p = 0.010$), and social functioning ($p < 0.001$). Regression analysis showed that longer walking distance significantly was associated with the RAND-36 physical functioning domain ($\beta = 6.45$, 95% confidence interval = 2.03–10.87, $p = 0.005$). People in the chronic phase post-stroke who had previously participated in LE- CIMT had reduced HRQoL compared to the general population regarding physical functioning, role-functioning physical, general health, and social functioning. A longer walking distance was associated with higher HRQoL in physical functioning domain, emphasising the importance of mobility training in post-stroke rehabilitation.

**Data availability statement:** The data cannot be shared publicly as it contains sensitive personal information about a relatively small number of individuals from a specific, localized area. This information could potentially be used to identify individuals, such as age, sex, time since stroke onset, time since treatment with LE-CIMT, affected side, living conditions, and physical functioning. According to the ethical approvals from the Swedish Ethical Review Authority in 2013 and 2020, access to the data is currently limited to participating researchers. Researchers may request de-identified data from the data owner, County Council of Västerbotten (Region Västerbotten), at fou. datauttag@regionvasterbotten.se, or contact the corresponding author for guidance. Access may require an approved application from the Swedish Ethical Review Authority (https://etik-provningsmyndigheten.se).

**Funding:** This study was supported by grants from the Centre for Clinical Research and Education, Karlstad, Region Värmland, Region Västerbotten, Umeå, Umeå University (ALF Foundation), the Swedish Stroke Foundation (Stroke Riksförbundet), and the Northern Swedish Stroke Fund (Strokeforskning i Norrland Insamlingsstiftelse ; https://www.strokeforskningnorrland.se/). The funders had no role in the study design, data collection and analysis, decision to publish, or preparation of the manuscript.

**Competing interests:** The authors have declared that no competing interests exist.

## Introduction

Stroke remains the third-leading cause of disability worldwide [1]. Every year, about 21,000 people in Sweden suffer a stroke, resulting in different disabilities [2]. The remaining disabilities may affect both their physical and cognitive functions and health-related quality of life (HRQoL) [3]. Independence in basic (bathing, toileting, indoor mobility, and dressing) and instrumental (shopping, driving, housekeeping, and food preparation) activities of daily living (ADL) has been shown to be associated with better HRQoL seven years post-stroke [3].

HRQoL is a holistic measure of an individual's perceived physical, mental, and social health [4]. Several instruments for assessing HRQoL are available, such as EuroQol Group five dimensions [5], The 36- Item Health Survey (version 1.0) distributed by RAND Corporation (RAND-36) [6,7], Medical Outcomes Study (MOS) Short Form-36 (SF-36) [8], and their variants. RAND-36 and MOS SF-36 use the same 36 items but differ in their scoring procedures for the eight domains, especially pain and general health. RAND-36 has been translated and culturally adapted into Swedish conditions and is free to use [9]. Recently, norm-based data became available [10].

Spontaneous recovery occurs early after a stroke, and rehabilitation is essential to speed up and strengthen recovery [11,12]. Even late after a stroke, it is possible to gain improvements due to the brain's plasticity and ability to reorganise [13,14]. Meta-analyses indicate that higher doses of task-specific training are necessary to improve motor functions and mobility after a stroke [15,16]. An intensive program of task-specific based therapy (i.e., constraint-induced movement therapy [CIMT]) six hours a day for two weeks has been effective in improving upper-extremity outcomes after a stroke [14,17,18]. Promising results for improving motor function, mobility and walking speed with lower-extremity CIMT (LE-CIMT) have recently been reported [19]. The results also indicated that patients treated with LE-CIMT had higher scores in basic ADL than those treated with conventional physiotherapy [20].

However, only a few studies have investigated HRQoL several years after a stroke [21,22]. One study evaluating day hospital rehabilitation after a stroke [23] showed significant improvements in HRQoL after the intervention, with improvements persisting at follow-up two years later [24]. In the meta-analysis by Zhou et al. [20], only three out of 34 studies evaluated quality of life (QoL) after LE-CIMT. Nonetheless, their results indicated higher QoL after LE-CIMT than conventional therapy. However, LE-CIMT requires some independent walking ability without walking aids before intervention [19] which indicate that participants who had received LE-CIMT already have a high level of motor function and balance pre-intervention. Although LE-CIMT seems to provide better conditions for higher HRQoL post-stroke, it is still unknown what difference there is in HRQoL after LE-CIMT compared to the general population and, if so, in which domains? In order to provide useful individualised rehabilitation interventions, more knowledge is needed regarding HRQoL after LE-CIMT compared to the general population and whether there are any determinants for better HRQoL.

This study aimed to investigate the HRQoL of people recovering from a stroke and who have participated in LE-CIMT and differences between sexes compared to the general population. Another aim was to evaluate whether various demographic characteristics and clinical outcomes are associated with HRQoL.

## Materials and methods

### Design and setting

This observational survey study was conducted among people who had experienced a stroke and had previously participated in LE-CIMT at an outpatient clinic in Stockholm, Sweden. It was approved by the Swedish Ethical Review Authority (Dnr. 2013–327-31M and amendment 2020–06189) and conformed to the Helsinki Declaration. The structure of this article complies with the Checklist for Reporting of Survey Studies [25].

### Questionnaire

We considered patient involvement to be important, and a questionnaire was prepared in collaboration with a representative from Neuro (https://neuro.se). Neuro is an association that focuses on making life easier for everyone with a neurological diagnosis. The questionnaire comprised open questions about age, stroke onset, and time since LE-CIMT and multi-choice questions about sex, ADL, living conditions, difficulties in cognitive functions, and HRQoL. The questionnaire started with the Swedish RAND-36 [9], including the Saltin–Grimby Physical Activity Level Scale (SGPALS) [26] and comprised 60 questions.

The Neuro representative, who had experienced a stroke, performed the pretesting and it took 11 minutes to answer the questionnaire. The representative lives in the same demographic area as the study population. The questionnaire is available in Swedish and can be requested from the corresponding author.

### RAND-36

The RAND-36 questionnaire is intended for use as a generic measure of perceived HRQoL. It comprises 36 items grouped into eight health domains: physical functioning, role- functioning physical, pain, general health, energy/fatigue, social functioning, role-functioning emotional, and emotional well-being. Each domain comprises 2–10 items. The score for the eight domains was calculated using the standard scoring algorithm by averaging 35 of the 36 items assessed with an ordinal scale. The last item asks about perceived health changes during the last year. Domain scores range from 0 to 100, with higher scores indicating better HRQoL [6,7]. The Swedish RAND-36 has been culturally adapted and tested for reliability and responsiveness [9]. The Mid-Health Survey has produced reference data for the Swedish RAND-36 that can be used for norm-based comparisons [10].

**Saltin-Grimby Physical Activity Level Scale (SGPALS).** SGPALS is a simple rating scale for physical activity and consists of one question about how much you move and exert physically during your leisure time. The questionnaire has mainly been used to assess physical activity levels in populations [27].

### Study sample and data collection

The inclusion criteria were people recovering from a stroke who had received six hours of LE- CIMT at an outpatient clinic in Stockholm. Patients were excluded if they had only participated in ≤ 3 hours of LE-CIMT.

Convenience sampling was used. In February 2021, 162 people who had participated in LE-CIMT at the clinic between 2001 and 2018 were sent an invitation, by regular mail, to participate along with an information letter, the questionnaire, and a prepaid envelope marked with an ID-code. They were allowed to get help from relatives when answering the questionnaire, and they could choose between answering the questionnaire on paper or by telephone. If they chose by telephone, the first author called them back, read them the questions, and registered their answers. After four weeks, a

thank-you/reminder letter was sent along with the questionnaire and a prepaid envelope to those who had not yet returned the questionnaire. After another six weeks, one last thank-you/reminder letter was sent along with the questionnaire and a prepaid envelope. The respondents provided written informed consent to participate in the study by answering the questionnaire. If responses were missing for parts of the questionnaire, the first author mailed the missing part back to the participant with a prepaid envelope or called them to complete the missing parts. The recruitment period for this study was from February 12 until June 6, 2021 and data were entered continuously and pseudonymised by the first author as the responses came in. After all responses were registered, the first author rechecked them. The code key was kept locked, and only the research team had access to it.

**Six-minute Walk test (6MWT).** The respondents results on 6MWT [28] post-LE-CIMT were retrospectively collected from our previous study [19]. To facilitate the interpretation of the result the 6MWT was converted into improvements in 100 metres increments.

## Statistical analysis

Descriptive characteristics are presented as mean ± standard deviation (SD) for continuous variables, and as number of cases (%) for categorical variables. A drop-out analysis compared age and sex between respondents and non-respondents using independent samples *t*-test and Chi-square test.

The scores for the eight RAND-36 domains, overall and by sexes, were compared between respondents and the norm-based data using a summary independent-samples *t*-test. The scores for the eight RAND-36 domains were compared, for the respondents, between sexes using the independent samples *t*-test.

Linear regression, adjusted for age and sex, was used to investigate how the 6MWT results, time since treatment, living alone status, and need for home care, respectively, were associated to RAND-36 domain scores. Histograms and quantile-quantile plots were used to verify the normal distribution assumptions for model residuals. For role-functioning physical and role-functioning emotional, normal distribution of the residuals could not be assumed, and ordinal proportional odds models were instead used.

There were no missing data for RAND-36, and a two-tailed *p*-value of <0.05 was considered statistically significant. Data were analysed using the Statistical Package for the Social Sciences for Windows (version 28.0; IBM, Chicago, IL, USA).

## Resvults

### Respondents' characteristics

One hundred six out of 162 people recovering from a stroke who previously had participated in LE-CIMT answered the questionnaire, a response rate of 65% (Fig 1). The respondents were aged 26–89 years, with a mean age of 62 ± 12 years. Sixty-five per cent were male, and 58% had right-sided hemiparesis. Ninety per cent of the respondents could move around indoors and outdoors independently, while 21% considered themselves physically inactive (SGPALS). When answering the questionnaire, 84% answered it themselves. The descriptive characteristics of the 106 respondents are summarised in Table 1. The 106 respondents and 56 non-respondents did not differ significantly in age (*p* = 0.37) or sex (*p* = 0.20) in the drop-out analysis.

### Health-related quality of life, RAND-36

The respondents estimated lower HRQoL than norm-based data in every domain except pain where it was equal [10]. The greatest differences were seen in the physical functioning and role-functioning physical domains (Fig 2).

The respondents had significantly decreased HRQoL overall and by sex (Table 2) for the physical functioning (total: *p* = 0.001; males; *p* = 0.001; females: *p* < 0.001), and role-functioning physical (total: *p* < 0.001; males: *p* < 0.001;

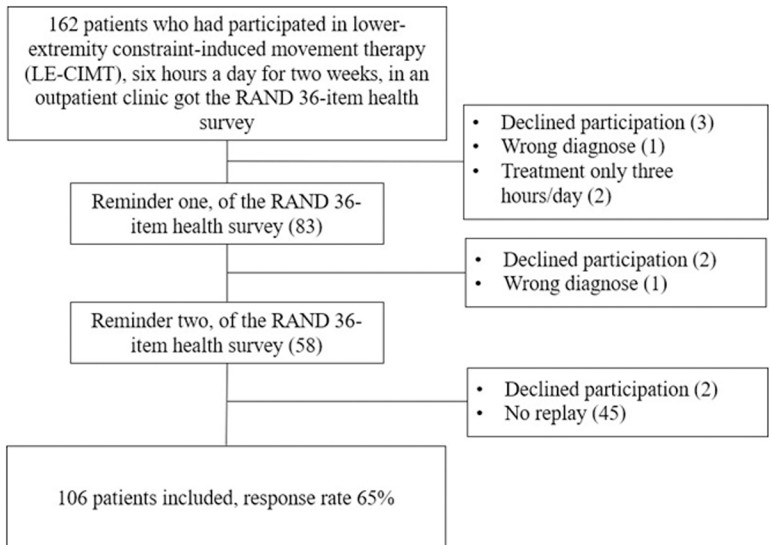

**Fig 1. Flowchart showing recruitment of the respondents.**

females: $p < 0.001$). The social functioning (total: $p < 0.001$; males: $p < 0.001$; females: $p = 0.052$), and general health domains (total: $p = 0.010$; males: $p = 0.022$; females: $p = 0.090$) showed significantly reduced HRQoL for the entire cohort and for males but not for females.

There were no significant differences in HRQoL for the pain (total: $p = 0.27$; males: $p = 0.23$; females: $p = 0.16$), energy/fatigue (total: $p = 0.89$; males: $p = 0.59$; females: $p = 0.61$); role-functioning emotional (total: $p = 0.39$; males: $p = 0.22$; females: $p = 0.31$), or emotional well-being (total: $p = 0.55$; males: $p = 0.15$; females: $p = 0.88$).

Male respondents rated higher self-assessed HRQoL in all RAND-36 domains compared to female respondents except for social functioning (66.5 and 72.6, respectively). However, the differences were not statistically significant: physical functioning ($p = 0.54$), role-functioning physical ($p = 0.60$), pain ($p = 0.47$), general health ($p = 0.43$), energy/fatigue ($p = 0.24$), social functioning ($p = 0.29$), role-functioning emotional ($p = 0.42$), and emotional well-being ($p = 0.64$).

Regarding health transition, 84% of the respondents experienced the same or better health in general now (at least two years post-stroke) compared to one year ago (Fig 3).

## Associations of health-related quality of life

Linear regression revealed a significant association between 6MWT per 100 metres and physical functioning ($\beta = 6.45$, 95% CI = 2.03–10.87, $p = 0.005$), i.e., an increase by 6.45 points in physical functioning is expected for every 100-m increase in 6MWT. No other significant associations were found (Table 3).

## Discussion

People recovering from a stroke who had previously undergone LE-CIMT had still significantly reduced perceived HRQoL in physical functioning, role-functioning physical, general health, and social functioning compared with Swedish norm-based data. However, we found no significant differences in pain, energy/fatigue, role-functioning emotional, and emotional well-being. Interestingly 6MWT distance was associated with higher HRQoL in physical functioning.

The result is consistent with studies from Norway and Finland that reported reduced perceived HRQoL among people recovering from a stroke in physical functioning, role-functioning physical, general health, and social functioning [29,30]. The Finnish study by Kauhanen et al. [30] found that depression and being marriage were significant and

**Table 1. Descriptive characteristics of study respondents (n = 106).**

| Age, mean ± SD (range), years | 62 ± 12 (26–89) |
|---|---|
| Time since stroke onset, mean ± SD (range), years | 10 ± 5 (3–33) |
| Time since treatment (LE-CIMT), mean ± SD (range), years | 8 ± 4 (2–20) |
| 6MWT after LE-CIMT, mean ± SD (range), metres | 389 ± 124 (82–715)[†] |
| Sex, male/female, *n* (%) | 69/37 (65/35) |
| More affected side, right/left, *n* (%) | 62/44 (58/42) |
| Own accommodation without domestic service, yes/no, *n* (%) | 96/10 (91/9) |
| Share household, yes/no, *n* (%) | 41/65 (39/61) |
| Answering the questionnaire, *n* (%)<br>　Themselves alone<br>　Themselves together with a relative<br>　Themselves by telephone | 89 (84)<br>13 (12)<br>1 (1) |
| By a relative | 3 (3) |
| Ability to move<br>　I can move around independently both indoors and outdoors<br>　I can move around independently indoors but not outdoors without help | 95 (90)<br>10 (9) |
| I need the help of another person both indoors and outdoors | 1 (1) |
| Physical activity and exercise (SGPALS), *n* (%)<br>　Some light physical activity<br>　Regular physical activity and training<br>　Regular hard physical training for competitive sports | 38 (36)<br>44 (41)<br>2 (2) |
| Physically inactive | 22 (21) |
| Have returned to gainful employment, *n* (%)<br>　No, I was not gainfully employed before suffering a stroke<br>　Yes, to the same extent as before the stroke<br>　Yes, but to a lesser extent than before the stroke | 8 (7.5)<br>13 (12)<br>23 (22) |
| No, but I plan to return to gainful employment | 8 (7.5) |
| No | 53 (50) |
| Do not know | 1 (1) |

*SD,* standard deviation; *LE-CIMT,* lower-extremity constraint-induced movement therapy; 6MWT, Six-Minute Walk Test; [†], 17 missing values; SGPALS, Saltin–Grimby Physical Activity Scale.

independent contributors to a low score in physical role limitations. Our study did not collect depression or marital status. However, role-functioning physical did not differ significantly between those living alone and those living with someone (Table 3).

In contrast, a study by Anderson et al. [22] following up on HRQoL 21 years after a stroke reported higher HRQoL scores than our study in all domains except for general health (equal) and energy/fatigue (lower). The stroke population, 21 years after stroke, did not differ from the general population and it might reflect that differences even out over time. The non-fatal disease burden has expanded globally with low back pain, headache disorders, and depressive disorders as the leading causes in both sexes combined [31]. This might explain why there was no difference in our study in the domains pain, fatigue, role-functioning emotional and emotional well-being. To understand whether there are any differences in, for example, the type of pain after stroke and pain in the general population, other instruments and study designs are required to explore this further.

Our cohort had higher scores in all eight RAND-36 domains than other stroke cohorts in the USA [32] and Sweden [33], especially in role-functioning physical (44.8), which was at least twice as high as the 25.1 reported in the USA cohort and 15.5 reported in the Swedish cohort. This may imply the high level of physical functioning in our LE-CIMT cohort. In addition, our cohort had higher scores in all eight RAND-36 domains than were reported in studies by

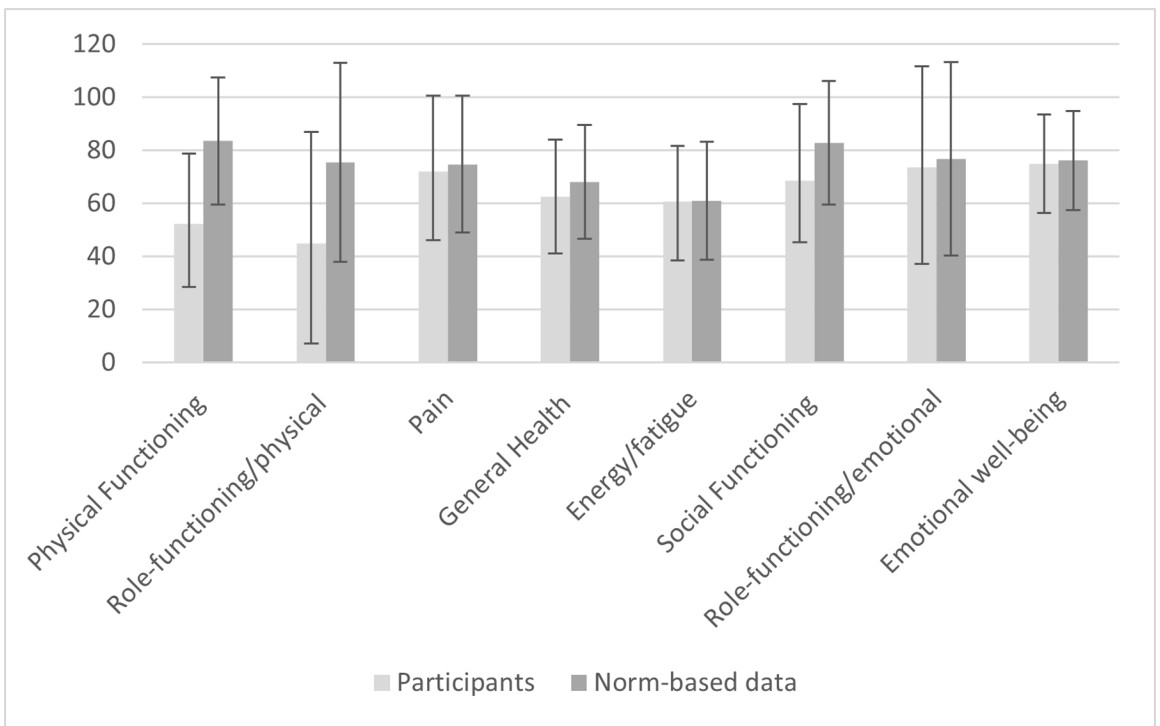

**Fig 2. Mean RAND-36 scale scores for the respondents versus norm-based data.** All respondents have had a stroke and rehabilitation with lower-extremity constraint- induced movement therapy (LE-CIMT). The norm-based data are drawn from the Mid-Swed Health Survey. Error bars showing standard deviation.

**Table 2. Mean (SD) RAND-36 scale scores in total and by gender for the respondents and reference data.**

| | Physical functioning | Role function- ing/ physical | Pain | General health | Energy/ fatigue | Social functioning | Role function- ing/ emotional | Emotional well-being |
|---|---|---|---|---|---|---|---|---|
| **Total** | | | | | | | | |
| Respondents | 52.4 (26.3) | 44.8 (42.1) | 71.9 (28.7) | 62.6 (21.3) | 60.7 (21.0) | 68.6 (28.8) | 73.6 (37.9) | 75.0 (18.5) |
| Reference | 83.5 (23.9) | 75.4 (37.6) | 74.7 (25.8) | 68.1 (21.5) | 61.0 (22.2) | 82.8 (23.4) | 76.7 (36.5) | 76.1 (18.6) |
| P-value[a] | **0.001**[*] | **<0.001**[*] | 0.27 | **0.010**[*] | 0.89 | **< 0.001**[*] | 0.39 | 0.55 |
| **Male** | | | | | | | | |
| Respondents | 53.5 (27.3) | 46.4 (42.5) | 73.4 (29.4) | 63.8 (22.1) | 62.5 (20.2) | 66.5 (29.3) | 75.8 (36.1) | 75.6 (18.9) |
| Reference | 85.6 (21.5) | 77.8 (34.3) | 76.3 (23.9) | 69.3 (19.4) | 63.6 (20.4) | 84.8 (21.4) | 79.9 (32.8) | 78.1 (17.3) |
| P-value[a] | **0.001**[*] | **<0.001**[*] | 0.33 | **0.022**[*] | 0.66 | **<0.001**[*] | 0.31 | 0.24 |
| **Female** | | | | | | | | |
| Respondents | 50.3 (24.6) | 41.9 (41.7) | 69.2 (27.7) | 60.4 (19.7) | 57.3 (22.4) | 72.6 (27.9) | 69.4 (41.1) | 73.8 (17.8) |
| Reference | 81.4 (26.0) | 73.1 (40.8) | 73.1 (27.5) | 66.9 (23.5) | 58.5 (23.7) | 80.8 (25.3) | 73.4 (39.8) | 74.1 (19.8) |
| P-value[a] | **<0.001**[*] | **<0.001**[*] | 0.39 | 0.09 | 0.76 | **0.052**[*] | 0.55 | 0.93 |

[a]Comparison of respondents to the reference data by Ohlsson-Nevo et al. from the Mid-Swed Health Survey (summary independent-samples t-test).

[*], a two- tailed p-value of ≤ 0.05 was considered statistically significant.

Olsson et al. [23,24] that evaluated day hospital rehabilitation after stroke and used the same inclusion criterion for age (18–65 years) as our study. Their role-functioning physical and role-functioning emotional scores (14.7 and 18.0) were less than half of ours (44.8 and 73.6). These differences may also be explained by the concepts of LE-CIMT

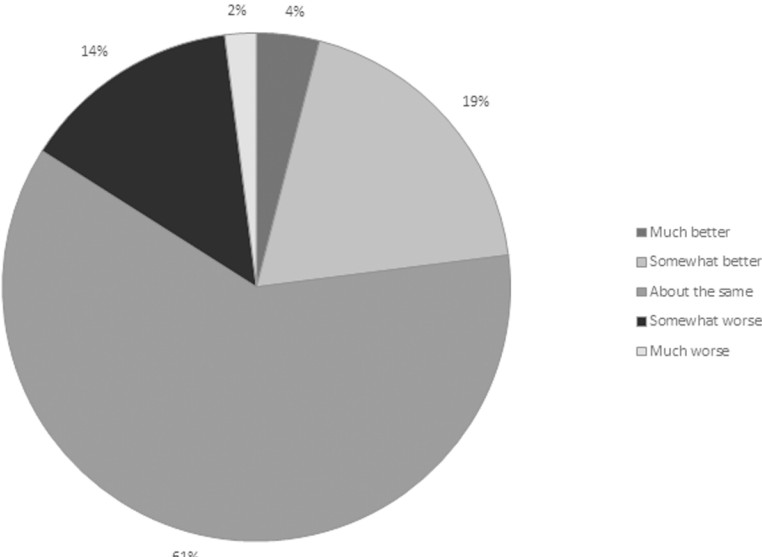

14%

2%   4%

19%

- Much better
- Somewhat better
- About the same
- Somewhat worse
- Much worse

61%

**Fig 3. Distribution of the respondents` present experience of health in general compared to one year ago.**

**Table 3. Univariable analysis of variance with a general linear model adjusting for age and sex for six of the eight RAND-36 domains or an ordinal model for the role-functioning physical and role-functioning emotional domains.**

| | Physical functioning | Role- function-ing physical[a] | Pain | General health | Energy/ fatigue | Social functioning | Role- function-ing emotional[a] | Emotional well-being |
|---|---|---|---|---|---|---|---|---|
| **6-minute walk test after treatment/100 meters** | | | | | | | | |
| β | 6.45 | 1.31 | -1.28 | 0.18 | -1.00 | 1.91 | 1.00 | -0.31 |
| 95% CI | 2.03-10.87 | 0.94-1.82 | -6.33-3.78 | -3.49-3.85 | -4.83-2.83 | -2.99-6.80 | 0.69-1.43 | -3.70-3.07 |
| p-value | **0.005*** | 0.11 | 0.62 | 0.92 | 0.60 | 0.44 | 0.99 | 0.85 |
| **Time since treatment** | | | | | | | | |
| β | -0.50 | 1.02 | -0.29 | -0.32 | -0.50 | -0.25 | 1.00 | 0.14 |
| 95% CI | -1.81- 0.81 | 0.93-1.12 | -1.81-1.23 | -1.44-0.81 | -1.60-0.60 | -1.77-1.28 | 0.90-1.10 | -0.84-1.12 |
| p-value | 0.45 | 0.69 | 0.70 | 0.58 | 0.37 | 0.75 | 0.94 | 0.78 |
| **Own accommodation with or without home care when answering the questionnaire** | | | | | | | | |
| β | 12.11 | 0.12 | -4.43 | 12.67 | -5.42 | 5.39 | 2.85 | 5.23 |
| 95% CI | -4.34-28.57 | -1.07-1.31 | -23.66-14.80 | -1.35-26.69 | -19.40-8.56 | -13.88-24.65 | 0.83-44.87 | -7.17-17.63 |
| p-value | 0.15 | 0.84 | 0.65 | 0.08 | 0.44 | 0.58 | 0.09 | 0.41 |
| **Live alone or share a household when answering the questionnaire** | | | | | | | | |
| β | -2.89 | -0.49 | -8.05 | -3.09 | 0.20 | -6.27 | 0.54 | -4.95 |
| 95% CI | -12.77-6.99 | -1.21-0.21 | -19.40-3.30 | -11.55-5.36 | -8.15-8.55 | -17.69-5.16 | 0.25-9.75 | -12.29-2.40 |
| p-value | 0.56 | 0.18 | 0.16 | 0.47 | 0.96 | 0.28 | 0.12 | 0.18 |

[a], ordinal regression; β, unstandardised beta; CI, confidence interval;

*, a two-tailed p-value of ≤ 0.05 was considered statistically significant.

requiring a high level of physical function and perhaps partly by the transfer package approach, where problem-solving is used to overcome perceived barriers to more affected leg use in ADL. However, the differences should be interpreted with caution due to the current study design, more robust designs are needed to draw conclusions about the efficacy of LE-CIMT.

Few studies have investigated HRQoL after LE-CIMT, and to our knowledge, none used the RAND-36. However, in the meta-analysis by Zhou et al. [20] the Stroke Specific Quality of Life scale and World Health Organization quality of life assessment were used and the results showed significant higher quality of life scores after LE-CIMT than conventional physiotherapy. This finding indicates that LE-CIMT may improve HRQoL more than other treatment methods. The main difference between LE-CIMT and other gait rehabilitations strategies is the intensity and the amount of training, six hours a day for two weeks. However, the LE-CIMT is performed accordingly to individually programs, with difficulty increasing with progress day by day. The patient has time to learn and practice over and over again how to perform everyday activities, potentially explaining the high RAND-36 scores in our study, even though still significantly reduced compared to the general population.

A significant positive relationship was found between the respondents' previous 6MWT distance and their HRQoL in physical functioning, with physical functioning domain scores increasing by 6.45 points for every 100-metre improvement in 6MWT distance. This result may be important for people recovering from a stroke since it is above the minimal clinically important difference of 3–5 points in the RAND-36 [34]. In a previous study [19], the 6MWT distance improved by ≥100 metres after LE-CIMT in 24 of the 147 participants. This finding has clinical relevance since physical functioning is often affected after a stroke, and that their HRQoL might be improved by increased walking ability through LE-CIMT even late after stroke.

## Strengths and limitations

Our study's strengths include the relatively large sample size with high response rate (65%) and the diversity in respondents' characteristics, that differed in age (26–89 years), time since LE-CIMT treatment (2–20 years), and 6MWT distances (82–715 metres), representing a broad spectrum of people recovering from a stroke. Another strength was the use of a well-known instrument (RAND-36) to assess HRQoL, culturally adapted and translated into Swedish and previously tested for reliability and responsiveness [9]. The Swedish RAND-36 was also used to collect the norm-based data, strengthening the reliability of the comparison between groups.

A limitation and a possible selection bias was that 56 out of 162 potential patients declined to participate. However, the drop-out analysis showed no significant difference in age or sex between respondents and non-respondents. It would have been a strength if physical functioning had been included in the dropout analysis. There is also a risk of selection bias when using convenience sampling because it is likely that the most motivated and resourceful people with moderate (and not severe) impairments will participate. Neither a control group nor assessments with HRQoL before LE-CIMT treatment were available for comparison, a limitation that affects external validity. Another factor that affects the external validity is that those who participate in LE-CIMT partly already have independent walking ability, albeit to a limited extent. Rather than generalising the findings to the entire stroke population, our study suggests that people in the chronic post-stroke stage who participated in LE-CIMT may have the same perceived HRQoL regarding pain, energy/fatigue, role-functioning emotional and emotional well-being as the general population in Sweden. This finding is consistent with the disability paradox already described by Albrecht and Devlieger in 1999 [35] and has to do with people establishing and maintaining a sense of balance in body, mind and spirit within their social context and environment despite the severity of their disability. Mavaddat et al. [36] replicated the study in 2021 among stroke survivors and confirmed that disability after stroke does not equate to poor health. Rehabilitation after stroke is multifaceted and requires knowledge of how different rehabilitation interventions together provide the conditions for living an easier life after a stroke. Further implementation of LE-CIMT into post-stroke rehabilitation is needed

Further research with longitudinal studies is needed to confirm the long-term impact of LE-CIMT after stroke. Including qualitative assessments and HRQoL to better understand which conditions need to be improved in the individual but also in society in order to reduce the difference in perceived HRQoL after stroke and the general population. It is vital to design interventions that could offer ways for stroke survivors to make sense of their predicament and increase their sense of control, confidence, independence, and self-determination in rehabilitation.

## Conclusions

Our study found significant differences in HRQoL between people recovering from a stroke who previously participated in LE-CIMT and the general population even though LE-CIMT might improve HRQoL. A longer walking distance was associated with higher HRQoL in physical functioning domain, indicating the importance of mobility training in post-stroke rehabilitation. However, more studies are needed to understand the complexity of post-stroke rehabilitation and how it affects people's HRQoL.

## Acknowledgments

We thank all the respondents who took the time to answer the survey and the Neuro Association for their participation in developing the questionnaire.

## Author contributions

**Conceptualization:** ingela marklund, Xiaolei Hu.

**Formal analysis:** ingela marklund, Per Liv, Xiaolei Hu.

**Funding acquisition:** ingela marklund, Xiaolei Hu.

**Investigation:** ingela marklund.

**Methodology:** Per Liv, Xiaolei Hu.

**Project administration:** ingela marklund.

**Supervision:** Brynjar Fure, Maria Klässbo, Britt-Marie Stålnacke, Xiaolei Hu.

**Visualization:** ingela marklund.

**Writing – original draft:** ingela marklund, Per Liv, Xiaolei Hu.

**Writing – review & editing:** Brynjar Fure, Maria Klässbo, Britt-Marie Stålnacke, Xiaolei Hu.

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
