## [Decision Letter · Decision Letter 0]

20 Dec 2024

PONE-D-24-53802Post-stroke health-related quality of life after lower-extremity constraint-induced movement therapy - A cross-sectional surveyPLOS ONE

Dear Dr. marklund,

Thank you for submitting your manuscript to PLOS ONE. After careful consideration, we feel that it has merit but does not fully meet PLOS ONE’s publication criteria as it currently stands. Therefore, we invite you to submit a revised version of the manuscript that addresses the points raised during the review process.

**ACADEMIC EDITOR: ** Thank you for submitting your manuscript to the Plos One. After a critical external peer review by experts in the field, I found that this manuscript has merit but needs to fully meet journal publication criteria as it currently stands. Therefore, we invite you to submit a revised version of the manuscript addressing the concerns the reviewers raised, specifically regarding study methodology and the clarity of your presentation. Please see the attached reviewer comments and details below.

We look forward to receiving your revised manuscript.

Kind regards,

Dr Redoy Ranjan, MBBS, MRCSEd, Ch.M., MS (CV&TS), FACS

Academic Editor

PLOS ONE

Journal Requirements:

Reviewers' comments:

Reviewer's Responses to Questions

**Comments to the Author**

1. Is the manuscript technically sound, and do the data support the conclusions?

Reviewer #1: Yes

Reviewer #2: No

Reviewer #3: Yes

Reviewer #4: Partly

2. Has the statistical analysis been performed appropriately and rigorously? 

Reviewer #1: Yes

Reviewer #2: Yes

Reviewer #3: Yes

Reviewer #4: Yes

3. Have the authors made all data underlying the findings in their manuscript fully available?

Reviewer #1: Yes

Reviewer #2: No

Reviewer #3: Yes

Reviewer #4: Yes

4. Is the manuscript presented in an intelligible fashion and written in standard English?

Reviewer #1: Yes

Reviewer #2: Yes

Reviewer #3: Yes

Reviewer #4: Yes

5. Review Comments to the Author

Reviewer #1: The manuscript is well detailed and the methods used is explicit . it is written in standard English as well.

can the topic be revised to "A cross-sectional evaluation of post-stroke health-related quality of life following lower-extremity 4 constraint-induced mobility therapy"

Reviewer #2: Please add MeSH related keywords.

"who had experienced a stroke and had previously participated in LE-CIMT" The methodology is not clear.

Why do you name the study as cross-sectional?

pre and post-procedure results should have been compared.

Some data were collected retrospectively.

Reviewer #3: The lack of pre-intervention HRQoL data prevents direct comparisons to baseline levels.

Potential selection bias due to non-respondents , should be discussed further.Explore additional interventions that complement LE-CIMT, such as cognitive and psychosocial support.

Include qualitative assessments to better understand the lived experiences of post-stroke patients undergoing LE-CIMT.

Highlight specific recommendations for integrating LE-CIMT into post-stroke rehabilitation protocols.

Emphasise the need for longitudinal studies to confirm the long-term impact of LE-CIMT.

Reviewer #4: Please see the attach for specific comments.THese comments should be shared with the author.

I responded partly for question 1 as some of their conclusions are overstated in the discussion given that this is an observational survey study.

6. PLOS authors have the option to publish the peer review history of their article (what does this mean? ). If published, this will include your full peer review and any attached files.

**Do you want your identity to be public for this peer review?** For information about this choice, including consent withdrawal, please see our Privacy Policy .

Reviewer #1: No

Reviewer #2: No

Reviewer #3: **Yes: ** Irma Ruslina Defi

Reviewer #4: No

---

## [Author Response · Author response to Decision Letter 1]

29 Jan 2025

Dear Editor

To the best of our ability, we have followed the PLOS ONE's style requirements and the figures has now been checked by PACE.

The ethical and legal restrictions on sharing our data set, has now been explained in detail.

The data cannot be shared publicly because it contains sensitive personal data on a relatively small number of individuals from a specified and local area which also can be used to identify people, e.g. age, sex, time since stroke onset, time since treatment with LE-CIMT, affected side, their living conditions and physical functioning. According to the ethical approvals, by the Swedish Ethical Review Authority, from 2013, and 2020, data access is currently limited to participating researchers. Researchers may request deidentified data from the data owner, County Council of Västerbotten (Region Västerbotten) fou.datauttag@regionvasterbotten.se or contact the corresponding author for guidance. Access may require an approved application from the Swedish Ethical Review Authority (https://etikprovningsmyndigheten.se).

Reviewer #1: The manuscript is well detailed and the methods used is explicit . it is written in standard English as well. Thanks

can the topic be revised to "A cross-sectional evaluation of post-stroke health-related quality of life following lower-extremity 4 constraint-induced mobility therapy" The topic is now revised.

Reviewer #2: Please add MeSH related keywords. In the PLOS one manuscript body formatting guidelines there is no information about MESH-related keywords at all. If it is ok with keyword I would like to add them.

"who had experienced a stroke and had previously participated in LE-CIMT" The methodology is not clear. Action taken. The sentence is clarified.

Why do you name the study as cross-sectional? Thank you for bringing this to our attention. The topic is now revised and also the method section.

pre and post-procedure results should have been compared. I agree, but it was not possible under my time as a PhD-student. I started my PhD-period in 2020 and the intervention with LE-CIMT ended 2018. I had only possibilities to collect post- intervention data of RAND-36.

Some data were collected retrospectively. Yes, their previously results on 6MWT were collected retrospectively. The sentence is now clarified in the method section.

Reviewer #3: The lack of pre-intervention HRQoL data prevents direct comparisons to baseline levels. Yes I agree, there is a lack and in forthcoming studies pre-intervention data should be included.

Potential selection bias due to non-respondents , should be discussed further. Action taken. I have now further developed the discussion around selection bias and non-responders

Explore additional interventions that complement LE-CIMT, such as cognitive and psychosocial support. Action taken. I have now further developed the discussion about additional interventions.

Include qualitative assessments to better understand the lived experiences of post-stroke patients undergoing LE-CIMT. I agree, and a qualitative study were performed earlier. ”I got knowledge of myself and my prospects for leading an easier life”: Stroke patients’ experience of training with lower-limb CIMT. https://www.tandfonline.com/doi/full/10.3109/14038190903141048

Highlight specific recommendations for integrating LE-CIMT into post-stroke rehabilitation protocols. Action taken. The discussion section has been revised.

Emphasise the need for longitudinal studies to confirm the long-term impact of LE-CIMT. Action taken. The discussion section has been revised.

Reviewer #4: Please see the attach for specific comments. These comments should be shared with the author. See table below, I have answered all the questions in the attached file

I responded partly for question 1 as some of their conclusions are overstated in the discussion given that this is an observational survey study. Action taken. The discussion section has been revised.

---

## [Decision Letter · Decision Letter 1]

20 Feb 2025

PONE-D-24-53802R1Post-stroke health-related quality of life following lower-extremity constraint-induced movement therapy - An observational survey studyPLOS ONE

Dear Dr. marklund,

Thank you for submitting your manuscript to PLOS ONE. After careful consideration, we feel that it has merit but does not fully meet PLOS ONE’s publication criteria as it currently stands. Therefore, we invite you to submit a revised version of the manuscript that addresses the points raised during the review process.

**ACADEMIC EDITOR: ** After a critical external peer review by the experts, I recommended a minor revision to improve the paper's clarity and presentation based on the reviewers' concerns. Please see the attached reviewer comments below.

We look forward to receiving your revised manuscript.

Kind regards,

Dr Redoy Ranjan, MBBS, MRCSEd, Ch.M., MS (CV&TS), FACS

Academic Editor

PLOS ONE

Journal Requirements:

Reviewers' comments:

Reviewer's Responses to Questions

**Comments to the Author**

1. If the authors have adequately addressed your comments raised in a previous round of review and you feel that this manuscript is now acceptable for publication, you may indicate that here to bypass the “Comments to the Author” section, enter your conflict of interest statement in the “Confidential to Editor” section, and submit your "Accept" recommendation.

Reviewer #1: All comments have been addressed

Reviewer #2: (No Response)

Reviewer #3: All comments have been addressed

Reviewer #4: All comments have been addressed

2. Is the manuscript technically sound, and do the data support the conclusions?

Reviewer #1: Yes

Reviewer #2: (No Response)

Reviewer #3: Yes

Reviewer #4: Yes

3. Has the statistical analysis been performed appropriately and rigorously? 

Reviewer #1: Yes

Reviewer #2: (No Response)

Reviewer #3: Yes

Reviewer #4: Yes

4. Have the authors made all data underlying the findings in their manuscript fully available?

Reviewer #1: Yes

Reviewer #2: (No Response)

Reviewer #3: Yes

Reviewer #4: Yes

5. Is the manuscript presented in an intelligible fashion and written in standard English?

Reviewer #1: Yes

Reviewer #2: (No Response)

Reviewer #3: Yes

Reviewer #4: Yes

6. Review Comments to the Author

Reviewer #1: (No Response)

Reviewer #2: Re-review of the manuscript: Post-stroke health-related quality of life following lower-extremity constraint-induced movement therapy - An observational survey study

Comments:

Some of the queries are not addressed properly.

Reviewer #3: The manuscript is well-structured and contributes valuable knowledge to post-stroke rehabilitation. Minor refinements in the discussion and methodological transparency would further strengthen its impact. I recommend acceptance with minor revisions.

Suggestion: 1. Consider discussing how LE-CIMT compares to other gait rehabilitation strategies in improving HRQoL.

2. Discussing why some HRQoL domains were not significantly different from the general population could enhance interpretation.

Reviewer #4: Thank you for the corrections. I did notice some typos in the new manuscript, in particular the discussion which should be corrected.

7. PLOS authors have the option to publish the peer review history of their article (what does this mean? ). If published, this will include your full peer review and any attached files.

**Do you want your identity to be public for this peer review?** For information about this choice, including consent withdrawal, please see our Privacy Policy .

Reviewer #1: No

Reviewer #2: No

Reviewer #3: **Yes: ** Irma Ruslina DEFI

Reviewer #4: No

---

## [Author Response · Author response to Decision Letter 2]

14 Mar 2025

Journal Requirements:

Thanks for bringing this to our attention. I have gone through the reference list and where the DOI has been missing I have inserted the PMID instead.

One reference Acarös Candan S, Livanelioglu A. Efficacy of modified constraint-induced movement therapy for lower extremity in patients with stroke: Strength and quality of life outcomes. Turk J Physiother Rehabil 2019;30(1):23-32. https://doi.org/10.21653/tfrd.406349. Has been replaced, I can´t find that the article has been retracted but the article is not searchable with PubMed so I chose to replace it anyway.

Reviewer #1: (No Response)

Reviewer #2: Re-review of the manuscript: Post-stroke health-related quality of life following lower-extremity constraint-induced movement therapy - An observational survey study

Comments:

Some of the queries are not addressed properly.

I am sorry, can you clarify which queries who are not addressed properly?

Reviewer #3: The manuscript is well-structured and contributes valuable knowledge to post-stroke rehabilitation. Minor refinements in the discussion and methodological transparency would further strengthen its impact. I recommend acceptance with minor revisions.

Suggestion: 1. Consider discussing how LE-CIMT compares to other gait rehabilitation strategies in improving HRQoL. Thanks, action taken.

I have now further developed the discussion around how LE-CIMT improving HRQoL compares to other gait rehabilitations strategies

2. Discussing why some HRQoL domains were not significantly different from the general population could enhance interpretation.

Thanks, action taken. I have now further developed the discussion around the domains who were not significantly different from the general population.

Reviewer #4: Thank you for the corrections. I did notice some typos in the new manuscript, in particular the discussion which should be corrected.

Thanks, I hope I found and correct all the typos you had notice

---

## [Decision Letter · Decision Letter 2]

6 Apr 2025

Post-stroke health-related quality of life following lower-extremity constraint-induced movement therapy - An observational survey study

PONE-D-24-53802R2

Dear Dr. marklund,

We’re pleased to inform you that your manuscript has been judged scientifically suitable for publication and will be formally accepted for publication once it meets all outstanding technical requirements.

Kind regards,

Dr Redoy Ranjan, MBBS, MRCSEd, Ch.M., MS (CV&TS), FACS

Academic Editor

PLOS ONE

Additional Editor Comments (optional):

Review Comments to the Author

Reviewer #3: The author has successfully addressed and accommodated all the comments I provided. I believe the revisions are sufficient and have adequately improved the manuscript. No further concerns remain from my side.

Reviewer #4: All comments have been addressed and is ready for submission by the journal.

Typos have been corrected.

---

## [Editor Report · Acceptance letter]

PONE-D-24-53802R2

PLOS ONE

Dear Dr. marklund,

I'm pleased to inform you that your manuscript has been deemed suitable for publication in PLOS ONE. Congratulations! Your manuscript is now being handed over to our production team.

Kind regards,

on behalf of

Dr. Redoy Ranjan

Academic Editor

PLOS ONE